# Feeding Pellets Containing Agro-Industrial Waste Enhances Feed Utilization and Rumen Functions in Thai Beef Cattle

**DOI:** 10.3390/ani13243861

**Published:** 2023-12-15

**Authors:** Natdanai Kanakai, Sawitree Wongtangtintharn, Chanon Suntara, Anusorn Cherdthong

**Affiliations:** Tropical Feed Resources Research and Development Center (TROFREC), Department of Animal Science, Faculty of Agriculture, Khon Kaen University, Khon Kaen 40002, Thailand; natdanai.k@kkumail.com (N.K.); sawiwo@kku.ac.th (S.W.); chansun@kku.ac.th (C.S.)

**Keywords:** beef, by-product, recycling, propionic acid, feed processing, citric fermented with yeast

## Abstract

**Simple Summary:**

This research addresses the issue of the limited nutritional value of citric waste and introduces the concept of citric waste fermented with yeast waste (CWYW) as a promising alternative nitrogen feed for cattle, boasting a high protein level of 53.5%. While prior studies have highlighted the benefits of CWYW, practical challenges related to its storage remain. In addition, this study explores adding CWYW pellets (CWYWP) to beef cattle feed at different amounts (0%, 2%, 4%, and 6% of their daily DM intake). It shows that adding 4% or 6% of DM per day can greatly increase beef cattle’s total CP intake as well as the CP digestibility, rumen bacterial population, and propionate concentration. Consequently, CWYWP presents a sustainable solution that reuses factory residue and aligns with the Bio-Circular-Green Economic Model (BCG) approach.

**Abstract:**

The objective of this research was to investigate the effects of citric waste fermented with yeast waste pellet (CWYWP) supplementation on feed intake, rumen characteristics, and blood metabolites in native Thai beef cattle that are fed a rice-straw-based diet. Four native male Thai beef cattle (1.0–1.5 years old) with an initial body weight (BW) of 116 ± 16 kg were held in a 4 × 4 Latin square design within 21-day periods. The animals were assigned to receive CWYWP supplementation at 0%, 2%, 4%, and 6% of the total dry matter (DM) intake per day. The results indicate that feeding beef cattle with CWYWP leads to a linear increase in the total intake as well was the intake of crude protein (CP) and the digestibility of CP, with the maximum levels observed at 6% CWYWP supplementation (*p* < 0.05). Rumen characteristics, including pH, blood urea-nitrogen concentration, and protozoal population, showed no significant alterations in response to the varying CWYWP dosages (*p* > 0.05). In addition, the CWYWP supplementation resulted in no significant changes in the concentration of ammonia-nitrogen, remaining within an average normal range of 10.19–10.38 mg/dL (*p* > 0.05). The inclusion of 6% CWYWP resulted in the highest population of ruminal bacteria (*p* < 0.05). Additionally, the CWYWP supplementation led to a statistically significant increase in the mean propionic acid concentration as compared to the group that did not receive the CWYWP supplementation (*p* < 0.05). In conclusion, this experiment demonstrates that supplementing Thai native beef cattle with CWYWP at either 4% or 6% DM per day can enhance their total CP intake as well as the CP digestibility and rumen bacterial population, and can increase propionate concentration.

## 1. Introduction

Climate change presents a global threat to livestock systems, affecting feed, water, animal health, and production due to global warming and shifts in climatic variables [1]. Additionally, it disrupts various aspects of cattle product supply chains, including processing, storage, transportation, retail, and consumption [2]. This imperils the capacity of current livestock systems to support livelihoods and meet the growing demand for animal products [3]. Consequently, farmers are seeking alternative protein sources and utilizing local feed resources to enhance production efficiency while reducing feed costs [4]. Yeast-derived protein has been identified as a promising dietary source capable of enhancing both rumen fermentation efficiency and beef cattle digestibility [5]. Yeast plays a vital role in the bioethanol industry, facilitating ethanol production through sugar fermentation [6]. Additionally, yeast waste is generated as a byproduct after ethanol production and has a significant protein content (CP) of 26.4% [7]. The research by Cherdthong et al. [8] demonstrated that yeast waste can fully replace soybean meal without adversely affecting feed intake, digestibility, or rumen fermentation.

Citric waste holds significant biological value and health benefits due to its rich contents of high-quality fiber, concentrated pectin, and various bioactive components [9]. Globally, citric waste production ranks as the second-largest fermentation process, yielding approximately 1.7 million tons annually, an amount that continues to grow [10]. *Aspergillus niger* exhibits fermentation capabilities when supplied with starch-rich source materials like rice, maize, and cassava [11]. Moreover, a substantial portion, up to 50–60% of the waste generated in citric acid production, can be repurposed [12]. Inadequate waste management, however, can lead to environmental pollution and related issues [13,14]. Citric waste comprises cellulose, hemicellulose, starch, and protein, making it suitable as animal feed [15]. A study conducted by Uriyapongson et al. [16] discovered that the inclusion of more than 10% of citric waste in a concentrated diet resulted in negative effects on nutrient intake and digestion in buffaloes. 

Furthermore, Suriyapha et al. [17] encountered limited nutritional value in citric waste, which prompted them to explore the creation of citric waste fermented with yeast waste (CWYW) as a cost-effective alternative nitrogen feedstuff for ruminants. The utilization of CWYW is a byproduct of obtaining citric waste from the agro-industrial production of citric acid, along with yeast waste generated by ethanol factories. This particular combination has a substantial CP content of 53.5%, indicating its potential as a significant protein source for ruminant diets. In their study, Suriyapha et al. [18] discovered that substituting 75% of soybean meal with CWYW in a concentrated diet did not result in any negative consequences for tropical cows or native beef cattle. However, practical issues related to CWYW powder storage may arise, which emphasizes the need to explore commercial market products like pellets for CWYW’s utilization.

Pelletized feeds are a widely adopted approach for the nutrition of ruminant animals, enabling them to consume complete diets and obtain essential nutrients efficiently [19]. These feed pellets are obtained by densifying ingredient mixtures into particle form, and it is crucial to assess their physical properties for proper design, management, and transportation systems [20]. Additionally, the use of pellets reduces feed waste, dust, and feed rejection by animals, effectively managing issues and diseases while maintaining diet quality [21,22]. Furthermore, feed pellets offer a more reliable and straightforward means of monitoring and delivering the ideal feed ratios to individual animals or groups that have specific nutritional requirements [19,20,23]. Many previous studies have shown that pellet feed can enhance nutritional digestibility and improve ruminal fermentation characteristics in ruminants [24]. One hypothesis for this is that the addition of CWYWPs could offer strategic enhancement, potentially improving rumen fermentation and the utilization of feed in animals. Consequently, this study aimed to explore the effects of the strategic supplementation of citric waste fermented with yeast waste pellets (CWYWPs) on feed intake, rumen characteristics, and blood metabolites in Thai native beef cattle fed on a rice-straw-based diet.

## 2. Materials and Methods

### 2.1. Ethical Procedure

The native Thai beef cattle utilized in this study were approved for use by the Animal Ethics Committee at Khon Kaen University, with the approval reference number IACUC-KKU-80/66 (issued on 20 July 2023).

### 2.2. Preparation of Citric Waste Fermented Yeast Waste Pellets 

A hundred liters of yeast waste (YW) were collected from an ethanol industry byproduct at Mitr Phol Bio-Power Co., Ltd., Chaiyaphum, Thailand. Additionally, 200 kg of citric waste (CW) were gathered from a byproduct of the citric acid business from Sam Mor Farm in Udon Thani, Thailand. Molasses (Molasses, Do Home Co., Ltd., Khon Kean, Thailand) and commercial-grade urea were purchased from a local shop. The formulation for the citric waste fermented with yeast waste pellets (CWYWPs) was adapted from Suriyapha et al. [17]. Initially, 100 L of YW were transferred to flask A. Then, in flask B, 20 kg of molasses and 50 kg of urea were mixed in 100 L of distilled water and blended. Solutions A and B were combined in a 1:1 ratio to create the YW media solution, followed by air-flushing the media solution for 16 h at room temperature using an electromagnetic air compressor (Sobo Mini AC/DC, Heiykungplatu Co., Ltd., Khon Kean, Thailand). Formic acid (Red Horse, Kd Rwmkaset Co., Ltd., Nong Khai, Thailand) was added to YW to adjust the pH range from 3.9 to 4.5. After 16 h, the YW media solution was transported and mixed with CW at 1000 L to 1000 kg ratio. The resulting product was placed in plastic bags (size 22 × 44 inches, PP Plastic Co., Ltd., Nakhon Ratchasima, Thailand) and vacuum-sealed (IMAFLEX 1400 W VS-921, Imarflex Industrial Co., Ltd., Bangkok, Thailand). Subsequently, it was stored in plastic bags under anaerobic conditions at room temperature for 14 days, followed by 48 h of exposure to the sun with a minimum moisture level of 10%. The CWYWP mixture was then subjected to sun-drying for 2–3 days (38–39 °C) to reduce its moisture content. Following this, it was mixed with distilled water to achieve a moisture level of 25%. The mixture was processed through a pelleting machine and allowed to sun-dry for an additional 2–3 days to reach a dry matter content of less than 10% before being incorporated into the animals’ diets [25].

### 2.3. Animals and Dietary Treatments

Four male Thai native beef cattle, aged between 1.0 and 1.5 years and having an initial body weight (BW) of 116 ± 16 kg, were selected for inclusion in a 4 × 4 Latin square design study spanning 21-day periods. These animals were allocated to different groups to receive CWYWP supplementation at varying levels: 0%, 2%, 4%, and 6% of the total DM intake each day. The CWYWPs were introduced as supplements to their diet at two daily intervals, specifically at 07:00 and 16:00. It is noteworthy that all of the Thai native bulls were individually housed in separate enclosures, and they had access to clean, fresh water as well as mineral blocks throughout the duration of the study.

Throughout the experiment, a concentrate diet with a CP content of 13.70% was administered daily at a rate ranging from 1.55% to 1.65% of the animal’s body weight (BW) in terms of dry matter (DM) intake. This feeding regimen aimed to achieve a daily weight gain of 0.50 kg, in accordance with the recommendations of the Working Committee of the Feeding Standard for Ruminants (WTRS) [26]. Subsequently, the bulls were provided with rice straw ad libitum [27]. The study encompassed four periods, each lasting 21 days. During the initial 14 days of each period, all animals received their designated treatments. For the final 7 days of each period, the cattle were transferred to metabolism crates to facilitate the collection of total fecal samples, which were analyzed to determine nutrient digestibility as follows:$$\mathrm{Digestibility}\;(\%)=\frac{\mathrm{Feed}\;\mathrm{Intake}\;\left(\mathrm{Kg}\;\mathrm{per}\;\mathrm{day}\right)-\mathrm{Fecal}\;\left(\mathrm{Kg}\;\mathrm{per}\;\mathrm{day}\right)}{\mathrm{Feed}\;\mathrm{Intake}\;\left(\mathrm{Kg}\;\mathrm{per}\;\mathrm{day}\right)}\times 100\%\;$$

In the last 7 days of the feeding trial, fecal samples were gathered using the metabolic cages designed for cattle. These specialized cages were situated within a spacious shed, each measuring 160 × 250 × 200 cm. They featured durable slatted floors with a trapezoid-shaped stainless-steel sheet beneath each cage. This design allowed for the floor to be positioned close to ground level, enabling the animals to take a single small step to enter. However, it also maintained sufficient elevation to accommodate the feces collection equipment underneath. This setup facilitated the easy collection and separation of feces, facilitated by a feces tray located at the back of each cage. Approximately 5% of the total fresh fecal samples were set aside, with the first part of each daily sample reserved for DM content analysis, while the second part of the daily samples was combined to create a single pooled sample. Detailed information regarding the ingredients in the concentrate and the nutrient concentrations of concentrate, rice straw, and CWYWPs can be found in Table 1.

The chemical compositions of the experimental diets are presented in Table 1. The concentrate diet contains cassava chips as the energy source, while the nitrogen sources are derived from soybean meal and urea. The concentrate feed contains a CP content of 13.7% on a dry matter basis, which is recommended to support beef cattle for maintenance when rice straw is provided as roughage. CWYWPs were developed as a protein supplement for beef cattle, boasting a CP content of 48.8% along with 36.6% and 24.8% for NDF and ADF contents, respectively. The elevated CP content offers additional nutrients, which is particularly beneficial when combined with low-quality roughage during the dry season. This high-protein value of CWYWP can be attributed to the production process, which involves the addition of 50% urea and 20% molasses to the yeast waste solution in CWYW. Moreover, the yeast cells present in the yeast waste might serve as a protein source because yeast provides amino acids and cell protein. Consequently, the development of pellet products could enhance protein utilization and feed efficiency when supplementing beef cattle.

### 2.4. Collection of Samples and Chemical Analysis

Throughout the feeding trial, the daily consumption of roughage, concentrate, and pellets by each animal was meticulously monitored by weighing both the offered and refused feed. Additionally, samples were collected twice a week to assess their DM contents and make necessary adjustments to the fresh weight of the concentrate diets. Fecal samples were consolidated into a single pooled sample, with each fecal sample weighing 500 g and frozen at −20 °C until the analysis during the last days of each trial period. In order to facilitate the chemical analysis, the diet samples underwent a series of preparation steps. Initially, the samples were thawed and underwent a drying process at a temperature of 60 °C for a duration of 72 h. Subsequently, the samples were finely ground through a 1 mm screen using a Cyclotech Mill manufactured by Tecator in Hoganas, Sweden. The chemical composition, encompassing dry matter (DM; ID 967.03), ash (ID 942.05), and crude protein (CP; ID 984.13), was determined in accordance with the AOAC standard method [28]. The amounts of neutral detergent fiber (NDF) and acid detergent fiber (ADF) in the samples were determined by boiling them in NDF and ADF solutions. Additionally, α-amylase was added, and the analysis followed the method developed by Van Soest et al. [29]. The intakes of concentrate, CWYWPs, rice straw, total dry matter, and nutrient intake were computed based on the aforementioned data collection. Ruminal fluid samples, with a volume of 100 mL each, were collected in a timely and efficient manner using a stomach tube connected to a vacuum pump. The samples were acquired at two time points: 0 h (before feeding) and 4 h after feeding. In order to avoid contamination with saliva, the first portion of the rumen fluid was thrown away, and the pH of the ruminal fluid was immediately measured using a pH meter (a HANNA Instruments HI 8424 microcomputer from Singapore). After the pH of the fluid samples was determined, they were separated into two distinct parts: the first 45 mL of fluid samples was placed in a plastic container with a capacity of 60 mL and given an addition of 9 mL of 1 M H_2_SO_4_ for the purpose of determining the concentrations of ammonia nitrogen (NH_3_-N) and volatile fatty acids (VFA), which included acetate (C2), propionate (C3), and butyrate (C4). The second sample of ruminal fluid, which was one milliliter and included nine milliliters of formalin containing 10% formaldehyde, was set aside for the counting of the number of bacterial and protozoal populations. After being collected, the samples of ruminal fluid were centrifuged at 16,000× *g* for 15 min, and then the supernatant was analyzed to assess the amounts of NH_3_-N and VFA present. The NH_3_-N concentration was quantified by employing the distillation technique and the Fawcett and Scott method [30]. The method used to determine the VFA concentration was gas chromatography (Agilent Technology, Santa Clara, CA, USA), and it adhered to the protocol that was outlined by Cherdthong et al. [8]. The enumeration of bacterial and protozoal populations was carried out using microscopic techniques with hemocytometers (Boeco, Hamburg, Germany) in accordance with the protocol described by Galyean et al. [31]. Blood samples were obtained from the jugular vein at the same time as rumen fluid collection, with an approximate volume of 10 mL. The samples were placed in tubes containing 12 mg of ethylene diamine tetraacetic acid (EDTA). Plasma was separated through centrifugation at 500× *g* for 10 min and subsequently preserved at −20 °C until the analysis for blood-urea-nitrogen (BUN) concentrations by using a diagnostic kit (L type Wako UN, Tokyo, Japan). The BUN is a generic name for plasma, serum, or whole-blood urea nitrogen. Following that, an automated colorimetric approach based on the diacetyl monoxime method [32] was used, which included a dialysis step to remove proteins for the analytical stream. A deproteinization step is necessary when this colorimetric approach is performed manually.

### 2.5. Statistical Analysis

The data analysis was performed using the General Linear Model (GLM) procedures in SAS [33], treating the experimental design as a 4 × 4 Latin square. Mean values and standard errors of means were used to present the results. Data were analyzed using the model:$$Y_{ijk}\;=\mu +M_{i}+A_{j}+P_{k}+\varepsilon_{ijk}$$
where $Y_{ijk,\;}$ is the observation from animal *j*, receiving diet *i*, in period *k*; *μ* is the overall mean, *M_i_* is the effect of the level of CWYWP (*i* = 1, 2, 3, 4), *A_j_* is the effect of animal (*j* = 1, 2, 3, 4); *P_k_* is the effect of the period (*k* = 1, 2, 3, 4); and *ε_ijk_* is the residual effect. To identify any significant differences among the treatments, Tukey’s Multiple Comparison Test was employed. Orthogonal polynomials were utilized to statistically compare the treatment trends. Ruminal ammonia-nitrogen concentrations were statistical analyzed using a repeated-measures-mixed-model to covariate for the initial values at time 0 h before feeding. The statistical significance was determined by differences among treatment means with *p*  <  0.05, signifying statistically significant distinctions.

## 3. Results

### 3.1. Feed Intake and Nutrient Intake

Table 2 displays the findings that concern the impact of CWYWP supplementation on the feed and nutrient intake in beef cattle. The results of this study indicate that providing CWYWPs to beef cattle did not influence their rice straw intake (*p* > 0.05), which remained within the range of 1.02 to 1.15 kg DM/day. Similarly, the addition of CWYWPs had no impact on the intake of organic matter (OM), ether extract (EE), neutral detergent fiber (NDF), and acid detergent fiber (ADF). Nevertheless, a significant linear increase in both the total intake and the intake of CP was observed with the increasing levels of CWYWP supplementation (*p* < 0.05).

### 3.2. Nutrient Digestibility

Table 3 presents the findings on CWYWP supplementation’s impact on nutrient digestibility. The influence of the supplementation levels of CWYWPs on the digestibility of DM, OM, EE, NDF, and ADF was found to be statistically insignificant *(p* > 0.05), with the exception of the digestibility of CP, which exhibited a significant alteration (*p* < 0.05). The addition of CWYWP supplementation has been seen to result in an increase in CP digestibility, with the maximum levels observed when 6% CWYWP supplementation was given.

### 3.3. Ruminal Ecology and Blood Urea Nitrogen

Table 4 presents the findings of the impact of CWYWP supplementation on rumen ecology and BUN levels. There were no significant effects observed on rumen pH, BUN concentration, or the protozoal population in response to varying dosages of CWYWPs (*p* > 0.05). Furthermore, the supplementation of CWYWP did not lead to any significant alterations in the concentration of NH_3_-N, which remained within the average-normal range of 10.19–10.38 mg/dL (*p* > 0.05). The inclusion of 6% CWYWP supplementation resulted in the greatest population of ruminal bacteria, with statistical significance at a significant level of *p* < 0.05.

### 3.4. Ruminal Volatile Fatty Acids (VFAs)

Table 5 presents the impact of CWYWP supplementation on the total concentration of the VFA and their respective profiles inside the rumen of native Thai beef cattle. The supplementation of CWYWPs in cattle did not have any significant impact on the levels of the total VFA and specific fatty acids such as acetic acid (C2), propionic acid (C3), and butyric acid (C4), or the ratio of acetic acid to propionic acid (C2 to C3 ratio) at both 0 and 4 h. Nevertheless, the inclusion of CWYWPs resulted in a statistically significant increase in the mean C3 concentration compared to the group that did not receive CWYWP supplementation (*p* < 0.05).

## 4. Discussion

### 4.1. Citric Waste Fermented with Yeast Waste Pellet (CWYWP)

In this study, the CP content of the CWYWPs was highest at 488 g/kg DM, which might be attributable to the fermentation process. This process involved adding yeast waste, urea, and molasses, leading to an increase in CP. In addition, the yeast waste containing live cell yeast, when combined with the urea and molasses, might supply substrate for yeast cell growth [34]. Therefore, when mixed with low-protein feedstuffs, yeast waste solution has the potential to increase the protein content of the final product [35]. Suriyapha et al. [18] observed that the CWYW contained a protein content of 535 g/kg DM, which is higher than the present work by an amount of 48 g/kg. This lower CP content of CWYWPs could be due to the pelleting process, in which some of the CP content may be lost due to high temperatures. However, the pelletizing method may be useful in encouraging feeding stability, extending storage time, preserving the nutritional content, and providing access to pellets with complementary nutritional qualities [36,37].

### 4.2. Feed Intake, Nutrient Intake, and Digestibility

Supplementing beef cattle with CWYWPs at 6% DM resulted in an increase in their total intake and their CP intake of up to 11.35% and 22.22%, respectively, as compared to the non-supplemented group. This increase could be attributed to the high CP content of CWYWPs (488 g/kg DM). As supplementation levels increased, a corresponding rise in the total intake and crude CP intake was observed. The higher CP intake is theorized as enhancing the availability of nitrogen, thereby serving as a crucial nitrogen source for microbial growth within the rumen [4,5]. This collaborative process works in conjunction with a carbon source, ultimately contributing to a more favorable microbial environment in the rumen. Therefore, this consumption has the potential to improve nutritional value and, in particular, to stimulate the growth of a variety of microorganisms in the rumen. These microorganisms include cellulolytic, amylolytic, and proteolytic bacteria, and the stimulation of their growth results in an increase in the digestibility of nutrients and the rates of digestion [38,39]. Ruminants have a distinct need for both true protein (bypass protein) and microbial crude protein (CP). Additionally, CWYWP supplementation presents a potential source of amino acids for ruminants. This is attributed to the pelleting process employed, which has the capacity to safeguard amino acid digestion from rumen microbes. As a result, the pelleting process facilitates the bypassing of amino acids through the rumen, allowing them to reach the small intestine where they can be absorbed and utilized by the animal [8,18]. The digestibility of crude protein increased by 3.81% in the animals supplemented with 6% CWYWPs as compared to the 0%-fed group. This increase could be attributed to the group with the highest level of CWYWP supplementation, which exhibited a high activity of microorganisms for digesting feed. The high microbial activity may be supported by CWYWPs, which provide a high-value nutritional supply for bacterial synthesis, resulting in improved feed digestion. As a result, the 6% CWYWP supplementation group showed the highest levels of bacterial population. This finding aligns with the results of Seankamsorn and Cherdthong [40], who reported that supplementation with 150 g/d of pellet feed containing 40.77% CP in native Thai beef cattle could increase CP digestibility by 10.65%. Furthermore, according to Matra and Wanapat [41], there is evidence to suggest that the inclusion of pellet feed with a CP content of 9.7% at a daily intake of 400 g can result in a 15.44% improvement in the digestibility of CP in dairy cows in comparison to cows fed with a daily intake of 300 g of pellet feed.

### 4.3. Ruminal Fermentation, Bacterial Number, and Blood Urea Nitrogen

The addition of CWYWP supplementation to the top-dress concentrate had no effect on the pH, with the cattle rumen pH ranging from 7.03 to 7.24 and 6.72 to 6.74 at 0 and 4 h after feeding, respectively. These pH ranges demonstrate the optimum rumen ecology for bacterial activity to break down the feed consumed and maintain rumen health [23,24]. In addition, when a suitable pH was obtained, the microbial cells could more effectively synthesize their cell growth by incorporating carbohydrates from a suitable source and a suitable level of NH_3_-N. Ruminal NH_3_-N concentration is a crude predictor of the efficiency of dietary N conversion into microbial N [42]. This present study demonstrated that the NH_3_-N concentrations did not change with varying levels of CWYWP supplementation. The lack of an increase in the rumen NH_3_-N, despite an escalating protein diet, could be attributed to rumen bacteria utilizing ammonia for their own cell synthesis. In the rumen, microorganisms, including bacteria, play a crucial role in fermenting and breaking down complex feed materials, such as CWYWPs [17]. When animals are fed a higher CWYWP diet, the increased availability of ammonia might lead to a greater microbial population in the rumen. Rumen bacteria utilize NH_3_ for protein synthesis, incorporating it into their own cells [18,42]. This microbial utilization of NH_3_ could explain why there was no proportional increase in the rumen NH_3_-N despite the rise in dietary protein. Moreover, the unique form of the feed supplement, which is presented as pellets with low degradation in the rumen, likely facilitates a slow-release nitrogen effect, ensuring that the rumen NH_3_-N concentrations did not increase when CWYWP supplementation was added [21,22]. In addition, rumen NH_3_-N has a high relationship with BUN concentration, which is another indicator of detected protein utilization efficiency [8]. The present study demonstrated that BUN did not change in animals fed the CWYWP group as compared to the non-supplemented group. This result could confirm that the nitrogen source in the rumen efficiently supplied nutrients for rumen microbial synthesis and did not lead to further urea generation in the blood [40].

The highest increase in the bacterial population was found when 6% of CWYWPs was supplemented. This can be explained by the hypothesis that, when sufficient essential nutrition is provided bacterial cells might synthesize more and increase their cell count in the rumen. The microbial population in the rumen might adapt to changes in the diet composition, optimizing the utilization of the available nitrogen without leading to an increase in ammonia levels. The findings of Matra and Wanapat [41] support the notion that incorporating pellet feed (9.7% CP) at a daily dosage of 400 g can lead to a 3% increase in the ruminal bacterial population in dairy cows, as compared to those fed with 300 g of pellet feed per day. In addition, Totakul et al. [43] demonstrated that the inclusion of supplementary pellets with a CP content of 23.6% in the diet of dairy bulls, at a daily consumption of 8% of dry matter, resulted in a significant increase of 26.31% in bacterial cell counts as compared to the group that did not receive the pellet supplementation.

### 4.4. Ruminal Volatile Fatty Acids (VFAs)

Propionate, generated within the rumen, serves as the primary precursor necessary for glucose synthesis in the liver, contributing to approximately 32% to 73% of the glucose required [44]. A recent experiment found that supplementing 4% and 6% CWYWP in beef cattle could increase the average concentration of ruminal propionate by 10.78% and 6.81% when compared to the 0% supplemented group which indicated CWYWP addition could be beneficial rumen efficiency. In line with the concept presented by Matra and Wanapat [41], feeding dairy cows 400 g of pellet feed per day (containing 9.7% CP) results in an 8.47% increase in ruminal propionate concentration as compared to feeding dairy cows only 300 g of pellet feed per day. According to the findings of Firkins and Mitchell [45], it was observed that ruminants on diets containing an appropriate amount of rumen-fermentable carbohydrates exhibited an elevation in the molar proportions of propionate. In this current study, it is probable that the observed increase in propionate content can be attributed to the existence of soluble carbohydrate constituents, such as molasses, in the CWYWPs [18]. The preparation of solution B for yeast activation, which included yeast waste, involved the addition of molasses at 20 g. This step was undertaken to facilitate yeast activation. Additionally, it is hypothesized that the peptidic fractions derived from *S. cerevisiae* in the yeast waste might have played a role in stimulating the growth of *Megasphaera elsdenii*. This bacterium is recognized as the primary lactate utilizer in the rumen, and it likely contributed to the conversion of lactate into propionate through the acrylate pathway [18]. Finally, the observation of concentrate intake in animals fed 4% and 6% of CWYWP supplementation showed that it was higher, at 2.21 to 3.39 g/kg BW^0.75^ respectively, as compared to 0% of CWYWP supplementation. Therefore, the higher concentrate intake may contribute to an additional substrate supply for propionate synthesis.

## 5. Conclusions

This experiment presents successful findings indicating that the addition of CWYWP to the diet of native Thai beef cattle, at either 4% or 6% dry matter per day, might result in improvements in their total intake, CP intake, ability to digest CP, and rumen bacterial population, and might also result in an increase in propionate concentration. The results of this study demonstrate that CWYWP may have the potential to provide a beneficial dietary supplement for enhancing the dietary requirements and performance of Thai native beef cattle.

## Figures and Tables

**Table 1 animals-13-03861-t001:** Dietary ingredients and chemical composition (±SD).

Feed Ingredient (% DM)	Concentrate Diet	CWYWP	Rice Straw
Cassava chips	50.0	-	-
Rice bran	19.0	-	-
Soybean meal	14.0	-	-
Palm kernel meal	14.0	-	-
Minerals and vitamins *	1.0	-	-
Salt	1.0	-	-
Urea	1.0	-	-
Chemical composition			
Dry matter, %	93.4 ± 0.50	94.9 ± 0.23	93.4 ± 0.35
Organic matter, % DM	92.6 ± 0.24	90.1 ± 0.21	88.8 ± 0.25
Crude protein, % DM	13.7 ± 1.34	48.8 ± 0.49	3.8 ± 0.86
Ether extract, % DM	3.9 ± 0.09	2.1 ± 0.08	0.7 ± 0.22
Neutral detergent fiber, % DM	18.9 ± 0.16	36.6 ± 1.31	72.3 ± 0.79
Acid detergent fiber, % DM	11.8 ± 0.14	24.8 ± 0.25	46.5 ± 0.54

* Vitamins and minerals: A: 10,000,000 IU; vitamin E: 70,000 IU; vitamin D: 1,600,000 IU; Fe: 50 g; Zn: 40 g; Mn: 40 g; Co: 0.1 g; Se: 0.1 g; I: 0.5 g.; CWYWP, citric fermented with yeast waste pellet.

**Table 2 animals-13-03861-t002:** Effect of citric waste fermented yeast waste pellet (CWYWP) supplementation on the feed intake of native Thai bulls.

Item	Level of CWYWP					*p* Values	Contrast		
	0%	2%	4%	6%	SEM	Treatment (T)	Control vs. T	Linear	Quadratic
Feed intake, kg dry matter/day									
Concentrate intake	2.26	2.27	2.31	2.38	0.38	0.94	0.59	0.61	0.87
CWYWP intake	0.00 ^c^	0.07 ^b^	0.14 ^a^	0.17 ^a^	0.02	<0.01	<0.01	<0.01	0.58
Rice straw intake	1.02	1.03	1.11	1.15	0.12	0.07	0.62	0.52	0.88
Total dry matter intake	3.28 ^b^	3.37 ^b^	3.56 ^ab^	3.70 ^a^	0.08	0.02	0.02	<0.01	0.67
Nutrient intake, kg/day									
Crude protein	0.35 ^c^	0.38 ^b^	0.43 ^a^	0.45 ^a^	0.01	<0.01	<0.01	<0.01	0.19
Organic matter	3.00	3.08	3.25	3.38	0.24	0.67	0.41	0.25	0.91
Ether extract	0.11	0.11	0.12	0.12	0.14	0.40	0.19	0.11	0.95
Neutral detergent fiber	1.16	1.20	1.29	1.34	0.17	0.88	0.55	0.45	0.98
Acid detergent fiber	0.74	0.76	0.82	0.86	0.12	0.89	0.61	0.47	0.64

^a,b,c^ Means with different letters in a row are significantly different at *p* < 0.05; SEM = standard error of the mean; Control (0% CWYWP) vs. T (2–6% CWYWP) = orthogonal contrast: control vs. level of CWYWP.

**Table 3 animals-13-03861-t003:** Effect of citric waste fermented yeast waste pellet (CWYWP) supplementation on the nutrient digestion of native Thai bulls.

Item	Level of CWYWP					*p* Values	Contrast		
	0%	2%	4%	6%	SEM	Treatment (T)	Control vs. T	Linear	Quadratic
Nutrient digestibility, %									
Dry matter	76.12	77.63	77.98	78.9	2.88	0.90	0.53	0.40	0.94
Organic matter	78.81	80.88	82.03	82.3	3.39	0.88	0.48	0.47	0.80
Crude protein	81.72 ^b^	83.36 ^ab^	83.48 ^ab^	84.96 ^a^	2.15	0.77	0.04	0.40	0.98
Ether extract	74.86	75.55	75.88	76.91	1.16	0.67	0.31	0.35	0.42
Neutral detergent fiber	73.78	73.7	75.97	75.59	1.89	0.76	0.57	0.40	0.95
Acid detergent fiber	72.87	73.63	74.72	73.89	2.43	0.96	0.51	0.56	0.62

^a,b^ Means with different letters in a row are significantly different at *p* < 0.05; SEM = standard error of the mean; Control (0% CWYWP) vs. T (2–6% CWYWP) = orthogonal contrast: control vs. level of CWYWP.

**Table 4 animals-13-03861-t004:** Effect of citric waste fermented yeast waste pellet supplementation on ruminal ecology and blood urea nitrogen.

Item	Level of CWYWP					*p* Values	Contrast		
	0%	2%	4%	6%	SEM	Treatment (T)	Control vs. T	Linear	Quadratic
Ruminal pH									
0 h	7.24	7.03	7.06	7.13	0.10	0.53	0.26	0.21	0.72
4 h	6.73	6.72	6.73	6.74	0.22	1.00	0.99	0.99	0.98
Mean	6.99	6.88	6.90	6.94	0.11	0.86	0.53	0.49	0.84
Ruminal ammonia-nitrogen, mg/dL									
0 h	9.90	9.82	9.87	9.81	0.03	0.20	0.10	0.14	0.77
4 h	10.83	10.64	10.51	10.94	0.13	0.20	0.41	0.77	0.06
Mean	10.37	10.24	10.19	10.38	0.07	0.25	0.26	0.95	0.06
Blood urea nitrogen (BUN), mg/dL									
0 h	11.25	11.50	10.50	11.50	0.55	0.57	0.90	0.92	0.52
4 h	11.50	13.00	12.50	12.00	0.56	0.35	0.17	0.70	0.12
Mean	11.38	12.25	11.50	11.75	0.45	0.56	0.41	0.86	0.51
Protozoal count, log10 cell/mL									
0 h	7.24	7.27	7.20	7.05	0.17	0.82	0.73	0.41	0.60
4 h	7.43	7.52	7.40	7.26	0.17	0.74	0.85	0.40	0.48
Mean	7.34	7.40	7.30	7.16	0.14	0.81	0.54	0.47	0.85
Bacterial count, log10 cell/mL									
0 h	8.48	8.49	8.65	8.70	0.06	0.18	0.85	0.40	0.38
4 h	8.79	8.81	8.87	8.93	0.22	0.72	0.34	0.29	0.87
Mean	8.64 ^b^	8.65 ^b^	8.76 ^ab^	8.82 ^a^	0.03	0.05	0.15	0.01	0.28

^a,b^ Means with different letters in a row are significantly different at *p* < 0.05; SEM = standard error of the mean; Control (0% CWYWP) vs. T (2–6% CWYWP) = orthogonal contrast: control vs. level of CWYWP.

**Table 5 animals-13-03861-t005:** Effect of citric waste fermented yeast waste pellet supplementation on volatile fatty acids (VFAs).

Item	Level of CWYWP				SEM	*p* Values	Contrast		
	0%	2%	4%	6%		Treatment (T)	Control vs. T	Linear	Quadratic
Total VFA, mM									
0 h	82.04	84.2	83.71	82.67	8.26	1.00	0.87	0.97	0.84
4 h	99.15	97.98	95.27	93.89	3.33	0.40	0.40	0.26	0.97
Mean	90.59	91.09	89.49	88.28	4.34	0.97	0.88	0.72	0.87
Acetic acid, mol/100 mol									
0 h	67.69	65.33	66.86	66.65	1.21	0.45	0.16	0.31	0.21
4 h	65.53	63.13	63.99	64.27	1.3	0.89	0.85	0.79	0.64
Mean	66.61	64.23	65.42	65.46	0.53	0.66	0.17	0.41	0.18
Propionic acid, mol/100 mol									
0 h	20.43	22.18	21.07	20.43	0.25	0.20	0.41	0.82	0.75
4 h	20.63 ^b^	21.53 ^ab^	24.95 ^a^	23.63 ^a^	0.25	0.01	0.23	0.73	0.02
Mean	20.53 ^b^	21.85 ^ab^	23.01 ^a^	22.03 ^a^	0.36	<0.01	<0.01	0.25	0.04
Butyric acid, mol/100 mol									
0 h	11.88	12.48	12.08	12.91	0.58	0.62	0.46	0.40	0.87
4 h	13.83	15.34	11.06	12.11	0.3	0.16	0.32	0.76	0.57
Mean	12.86	13.91	11.57	12.51	0.08	<0.01	<0.01	0.64	<0.01
Acetic acid to propionic acid ratio									
0 h	3.31	2.95	3.17	3.26	0.3	0.49	0.19	0.40	0.21
4 h	3.18	2.93	2.56	2.72	0.27	0.43	0.15	0.50	0.13
Mean	3.24	2.94	2.87	2.99	0.06	0.57	0.14	0.62	0.10

^a,b^ Means with different letters in a row are significantly different at *p* < 0.05; SEM = standard error of the mean; Control (0% CWYWP) vs. T (2–6% CWYWP) = orthogonal contrast: control vs. level of CWYWP.

## Data Availability

The data presented in this study are available on request from the corresponding author.

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
