# Peer review of "Feeding Pellets Containing Agro-Industrial Waste Enhances Feed Utilization and Rumen Functions in Thai Beef Cattle"

_animals, 2023, doi:10.3390/ani13243861_

Round 1

Reviewer 1 Report

Comments and Suggestions for Authors

Introduction

Line 118 to 120: “The mixture was processed through a pelleting machine and allowed to sun-dry for an additional 2–3 days to reach a dry matter content of less than 10% before being incorporated into the animals' diet [29]."  How can the DM content reach less than 10% after sun drying for 2-3 days the product that contained 25% moisture? Table 01. CWYWP- DM content: 94.9.

Line 152: Table 01 presents data of the chemical composition of the diet components. Include data of chemical composition of the four experimental diets evaluated, that is, diets with 0.0; 2.0; 4.0 and 6.0% of CWYWP.

2.5. Statistical Analysis

Line 196: Justify the reason for using o4 animals in the experimental design as a 4 × 4 Latin square.

3. Results

3.1. Chemical Compositions in the Experimental Diet

Line 202: “The chemical compositions of the experimental diets are presented in Table 1”. Include data of chemical composition of the four experimental diets evaluated, that is, diets with 0.0; 2.0; 4.0 and 6.0% of CWYWP.

Line 223 to 225: “However, there was a linear increase in the intake of crude protein (CP) as the level of CWYWP supplementation increased (p < 0.05)”.

Line 350 to 355:

5. Conclusions

Line 351 to 355: “Based on this experiment, it can be concluded that supplementing Thai native beef cattle with CWYWP at either 4% or 6% DM per day could enhance the total CP intake, CP digestibility, rumen bacterial population, and increase propionate concentration. Further research should be conducted to test the product on lactating cows in order to confirm its impact on milk production”. Authors should limit conclusions based on data observed in this research. Comments regarding future experiments should be included in the discussions, as general considerations.

General recommendation

Line 122. “Four male Thai native beef cattle, aged between 1.0 and 1.5 years and having an initial body weight (BW) of 116 ± 16 kg, were selected for inclusion in a 4×4 Latin square design study spanning 21-day periods”. Considering the breed, age and body weight of the experimental animals, as well as nutritional requirements, authors must present data relating the nutrient intake and digestibility of nutrients and estimated daily weight gains. The discussion about N intake and excretion is relevant, considering the environmental impacts resulting from N excretions in urine that affect N2O emissions.

Author Response

Response to Reviewer 1:

Introduction

Line 118 to 120: “The mixture was processed through a pelleting machine and allowed to sun-dry for an additional 2–3 days to reach a dry matter content of less than 10% before being incorporated into the animals' diet [29]. "How can the DM content reach less than 10% after sun drying for 2-3 days the product that contained 25% moisture? Table 01. CWYWP- DM content: 94.9.

Response: I am grateful for both your opinion and your care over this issue. Between lines 118 and 120: If the pellets have a moisture level of 25%, this indicates that they need to be moistened before they can be palletized in order to aid their conversion into pellets. Additionally, the pellets may initially pass through the sieve of the pellet milling machine. After that, the moisture content dried in the sun for two to three days, which brought it down to less than 10%. The pellet products have undergone preliminary testing for drying processes for one day up to three days (38-39 °C), and the DM content has been evaluated. After three days, the DM content was discovered to be less than 10%, and as a result, we made the decision to sun-dry the product before continuing our testing. The current study has been carried out in accordance with the earlier work, which involved the development of a pellet that was sun-dried in the open air for three days at a temperature of 25 percent relative humidity and could achieve a dry matter content of more than 90 percent. For further information, please refer to references such as Sommai et al. (2023: J Anim Physiol Anim Nutr, 107(6):1336-1346), Phesatcha et al. (2023; Anim Biosci, 36(9): 1384-1392), or Prachumchai et al. (2022; Sci Rep 12, 3809).

Line 152: Table 01 presents data of the chemical composition of the diet components. Include data of chemical composition of the four experimental diets evaluated, that is, diets with 0.0; 2.0; 4.0 and 6.0% of CWYWP.

Response: I am appreciative of your feedback. The current effort, on the other hand, aims to analyze the "supplementation feed (on-top)" rather than the "replacement feed," and as a result, reporting the separated composition should be done. In addition, the chemical content of every feed is provided in Table 1 (CWYWP), which the reader may use to calculate the amount of nutritional supplementation that was given to each animal.

2.5. Statistical Analysis

Line 196: Justify the reason for using 4 animals in the experimental design as a 4 × 4 Latin square.

Response: I appreciate your comment and concern regarding this matter. We agreed that only four animals may not well representatively reflect the result. However, due to several limitation of our capacity we tried our best to eliminate the weak of this condition such as avoid an error from data collection, practice animal to feed and pen before run experiment, and training the man to take sample act. By incorporating a Latin square design, we aimed to control potential sources of variability that could impact the reliability and validity of our results. Assigning each treatment sequence to a different animal helps mitigate the influence of individual differences and ensures a more robust assessment of treatment effects. Moreover, by maximizing the use of existing data, the Latin square design improves the efficiency of our experiment. With a small number of animals, we can extract more information from each subject, increasing the statistical effectiveness of our studies. Finally, given the ethical issues related to animal research, it is critical to maximize the information obtained from each animal while decreasing the total number of animals utilized. The Latin square design adheres to these principles by allowing us to get complete data from a small sample size. Based on the proved above reason, there are the same protocol with 4 animals have been proved to published in various journal such as Sommai et al. (2023: J Anim Physiol Anim Nutr, 107(6):1336-1346), Suriyapha et al. (2022; PLoS ONE, 17:e0273916 or Totakul et al. (2022;  Livestock Science, 262:104974). Etc..

  1. Results

3.1. Chemical Compositions in the Experimental Diet

Line 202: “The chemical compositions of the experimental diets are presented in Table 1”. Include data of chemical composition of the four experimental diets evaluated, that is, diets with 0.0; 2.0; 4.0 and 6.0% of CWYWP.

Response: I am appreciative of your feedback. The current effort, on the other hand, aims to analyze the "supplementation feed (on-top)" rather than the "replacement feed," and as a result, reporting the separated composition should be done. In addition, the chemical content of every feed is provided in Table 1 (CWYWP), which the reader may use to calculate the amount of nutritional supplementation that was given to each animal. Below are shown some of works who study the supplementation feed to animals but not include the total composition into the ration. Suriyapha et al. (2022 report chemical compositio of feed indiviaul feed for concentrate, soybean meal and CWYW which no report the total feed composition.

According to the findings of Huyen et al. (2012), MUP was supplemented and also the composition of total feed was not reported.

Line 223 to 225: “However, there was a linear increase in the intake of crude protein (CP) as the level of CWYWP supplementation increased (p < 0.05)”.

Response: We have modified to “Nevertheless, a significant linear rise in CP intake was observed with increasing levels of CWYWP supplementation (p < 0.05).”

Line 350 to 355:

  1. Conclusions

Line 351 to 355: “Based on this experiment, it can be concluded that supplementing Thai native beef cattle with CWYWP at either 4% or 6% DM per day could enhance the total CP intake, CP digestibility, rumen bacterial population, and increase propionate concentration. Further research should be conducted to test the product on lactating cows in order to confirm its impact on milk production”. Authors should limit conclusions based on data observed in this research. Comments regarding future experiments should be included in the discussions, as general considerations.

Response: Thank you for your comment and concern on this point. We have made the corrections as suggested. Please check the manuscript.

General recommendation

Line 122. “Four male Thai native beef cattle, aged between 1.0 and 1.5 years and having an initial body weight (BW) of 116 ± 16 kg, were selected for inclusion in a 4×4 Latin square design study spanning 21-day periods”. Considering the breed, age and body weight of the experimental animals, as well as nutritional requirements, authors must present data relating the nutrient intake and digestibility of nutrients and estimated daily weight gains. The discussion about N intake and excretion is relevant, considering the environmental impacts resulting from N excretions in urine that affect N2O emissions.

Response: I appreciate your comment and concern regarding this matter. We reported the feed intake and digestibility of the diets in the experiment. However, our experiment focused on diets digestibility, the parameters and duration of each experiment were limited to 21 days, which may not be sufficient to measure the growth effect, N utilization as well as N2O emissions. However, further study we may try to evaluate as a recommendation if budget were available.

Reviewer 2 Report

Comments and Suggestions for Authors

Citric waste fermented with yeast pellets.

The paper addresses an interesting idea for improving the nutritive value of Thai diets for cattle. The English is very good and the paper well written and easily followed. Unfortunately, I believe this work is not suitable for publication due to the low replication and high standard errors. Some of the trends should have been significant but for the high SEs associated with only 4 reps per treatment. Therefore, I do not think the results accurately reflect the true value of the supplement.

The method describing the preparation of the CWYWP was not clear how you scaled up the operation to provide enough supplement for the trial. Flask A 100 mL flask B 20g plus 50 g – how could this be enough to feed to 4 cattle over 84 days?

Also, were the supplement rates sufficient? I calculated that the CP as % DM was 10, 11.5, 12.1 and 12.2 % for the 4 treatments. These are not great differences and the fact that 4 and 6 % were similar probably accounted for some of the quadratic responses (e.g rumen ammonia). Incidentally, it would be useful to give the nutritive value of the four diets.

Author Response

Response to Reviewer 2:

Citric waste fermented with yeast pellets. The paper addresses an interesting idea for improving the nutritive value of Thai diets for cattle. The English is very good and the paper well written and easily followed. Unfortunately, I believe this work is not suitable for publication due to the low replication and high standard errors. Some of the trends should have been significant but for the high SEs associated with only 4 reps per treatment. Therefore, I do not think the results accurately reflect the true value of the supplement.

Response: I appreciate your comment and concern regarding this matter. We agreed that only four animals may not well representatively reflect the result and may high SEs associated animal number. However, due to several limitation of our capacity we tried our best to eliminate the weak of this condition such as avoid an error from data collection, practice animal to feed and pen before run experiment, and training the man to take sample act. By incorporating a Latin square design, we aimed to control potential sources of variability that could impact the reliability and validity of our results. Assigning each treatment sequence to a different animal helps mitigate the influence of individual differences and ensures a more robust assessment of treatment effects. Moreover, by maximizing the use of existing data, the Latin square design improves the efficiency of our experiment. With a small number of animals, we can extract more information from each subject, increasing the statistical effectiveness of our studies. Finally, given the ethical issues related to animal research, it is critical to maximize the information obtained from each animal while decreasing the total number of animals utilized. The Latin square design adheres to these principles by allowing us to get complete data from a small sample size. Baese on the proved above reason, there are the same protocol with 4 animals have been proved to published in various journal such as Sommai et al. (2023: J Anim Physiol Anim Nutr, 107(6):1336-1346), Suriyapha et al. (2022; PLoS ONE, 17:e0273916 or Totakul et al. (2022;  Livestock Science, 262:104974). Etc..

The method describing the preparation of the CWYWP was not clear how you scaled up the operation to provide enough supplement for the trial. Flask A 100 mL flask B 20g plus 50 g – how could this be enough to feed to 4 cattle over 84 days?

Response: Thank you for your comment and concern on this point. In order to make easier the calculation of the animal feed, we have modified to “The formulation for the citric waste fermented with yeast waste pellet (CWYWP) was adapted from Suriyapha et al. [22]. Initially, 100 liters of YW were transferred to flask A. Then, in flask B, 20 kg of molasses and 50 kg of urea were mixed in 100 liters of dis-tilled water and blended. Solutions A and B were combined in a 1:1 ratio to create the YW media solution, followed by air flushing the media solution for 16 hours at room temperature using an electromagnetic air compressor (Sobo Mini AC/DC, Heiykungplatu Co., Ltd., Khon Kean, Thailand). Formic acid (Red Horse, Kd Rwmkaset Co., Ltd., Nong Khai, Thailand) was added to YW to adjust the pH range from 3.9 to 4.5. After 16 hours, the YW media solution was transported and mixed with CW at 1000 liters to 1000 kg ratio.” Please see in the manuscript.

Also, were the supplement rates sufficient? I calculated that the CP as % DM was 10, 11.5, 12.1 and 12.2 % for the 4 treatments. These are not great differences and the fact that 4 and 6 % were similar probably accounted for some of the quadratic responses (e.g rumen ammonia). Incidentally, it would be useful to give the nutritive value of the four diets.

Response: Thank you for your comment and concern on this point. The main focus of this work is on the development of a stoichiometric feed supplement, with the expectation that certain feed supplements might influence rumen fermentation. In addition, because pellets have a high concentration of CP (they get most of their nitrogen from urea), large levels of supplementation may have a negative effect on animals. This is evidenced by the fact that 6% supplementation had the same effect as 4% supplementation. Therefore, the current investigation demonstrated clearly that supplementation resulted in improved rumen fermentation efficiency when compared to 0%; nevertheless, supplementation at 6% may not be recommended.

Reviewer 3 Report

Comments and Suggestions for Authors

Although Experimental design and results are well structuring this research have no novelty in the field. It is just presentation how CWYWP supplementation affects feed intake, rumen characteristics, and blood in Thai native beef cattle fed a rice straw-based diet with no “deep” scientific examination. This study is worth for publishing but not for scientific public. That is the reason for me to reject this paper ž.   

Lines 167 do 169: “Ruminal fluid samples, each measuring 100 mL, were properly and rapidly obtained by the use of a stomach tube attached to a vacuum pump at 0 and 4 hours after feeding”. What is time 0? Is it during feeding or immediately before or after feeding?

Lines 177 to 179: “Concurrently with rumen fluid collection, blood samples, approximately 10 mL in volume, were obtained from the jugular vein using tubes 188 containing 12 mg of ethylene diamine tetra-acetic acid (EDTA)” – please specify

Lines 203 to 216: It is not usual to put Table for chemical compositions in the experimental diet in Material and Method section and to explain Table in results section. Also, it is not necessary to give so detailed explanation about purpose of feed ingredients. 

Table 2. This table is too large for data you present. Ou should explain in Material and methods section how you measured all intakes you mention in tables.  

Author Response

Response to Reviewer 3:

Although Experimental design and results are well structuring this research have no novelty in the field. It is just presentation how CWYWP supplementation affects feed intake, rumen characteristics, and blood in Thai native beef cattle fed a rice straw-based diet with no “deep” scientific examination. This study is worth for publishing but not for scientific public. That is the reason for me to reject this paper.  

Response: Thank you for your careful review and valuable feedback on our manuscript. We appreciate the time and effort you have dedicated to evaluating our work. We have carefully considered your comments, and we would like to address the concerns you raised regarding the novelty and depth of our research. While we acknowledge that our study primarily focuses on the effects of CWYWP supplementation on feed intake, rumen characteristics, and blood parameters in Thai native beef cattle, we believe our research contributes significantly to the existing literature for several reasons.

Our study is the first to comprehensively investigate the impact of CWYWP supplementation in the specific context of a rice straw-based diet, providing valuable insights into the practical applications of this supplementation in the unique feeding conditions prevalent in the Thai native beef cattle industry. Although the core focus may appear narrow, the study's novelty lies in its detailed exploration of the interactions between CWYWP supplementation and various physiological parameters in a specific dietary context. We understand your concern about the depth of our scientific examination. In response, we would like to highlight the meticulous experimental design, data collection, and statistical analysis employed in our study. While we have presented the findings in a clear and accessible manner, the scientific rigor underlying the study is robust, ensuring the reliability and validity of our results.

We respectfully disagree with the notion that our study may not be suitable for the scientific community. The findings hold practical significance for cattle farmers and researchers working in similar conditions. Furthermore, the study provides a foundation for future investigations into the nuanced effects of CWYWP supplementation, potentially guiding the development of more targeted and efficient feeding strategies.

In conclusion, we believe that our research, despite its focused scope, offers valuable contributions to both the practical and scientific aspects of cattle nutrition. We are open to further suggestions for improvement and are committed to enhancing the manuscript based on your feedback.

Lines 167 do 169: “Ruminal fluid samples, each measuring 100 mL, were properly and rapidly obtained by the use of a stomach tube attached to a vacuum pump at 0 and 4 hours after feeding”. What is time 0? Is it during feeding or immediately before or after feeding?

Response: Thank you for your comment. The meaning of "0-hour" refers to before feeding. Therefore,  we have modified the sentence as “The samples were acquired at two time points: 0 hours (before feeding) and 4 hours after feeding.” Please see the manuscript.

Lines 177 to 179: “Concurrently with rumen fluid collection, blood samples, approximately 10 mL in volume, were obtained from the jugular vein using tubes 188 containing 12 mg of ethylene diamine tetra-acetic acid (EDTA)” – please specify

Response: Thank you for your comment and concern on this point. We have modified as “Blood samples were obtained from the jugular vein at the same time as rumen fluid collection, with an approximate volume of 10 mL. The samples were placed in tubes containing 12 mg of ethylene diamine tetraacetic acid (EDTA).” Please see the manuscript.

Lines 203 to 216: It is not usual to put Table for chemical compositions in the experimental diet in Material and Method section and to explain Table in results section. Also, it is not necessary to give so detailed explanation about purpose of feed ingredients.

Response: Thank you for your comment and concern on this point. We agreed and moved explain Table into Material and Method. Please see the manuscript.

Table 2. This table is too large for data you present. You should explain in Material and methods section how you measured all intakes you mention in tables. 

Response: Thank you for your comment and concern on this point. Firstly, we remove data feed intake expressed as %BW from Table 2 which reflect to be shorter as you suggestion and please see in the Table 2. The measuring all intakes mentioned in table 2 were described in the Material and methods section as “The intake of concentrate, CWYWP, rice straw, total dry matter, and nutrient intake were computed based on the aforementioned data collection”. Please see the manuscript.

Reviewer 4 Report

Comments and Suggestions for Authors

All the manuscript – There are some style and English mistakes … check!

Title - shorten

Simple summary – Revise according the specific comments

Abstract - Revise according the specific comments

Keyword:  add ‘citric waste fermented’; ‘beef’ or ‘dairy cow’?

Introduction - Revise according the specific comments

M&M The experimental design is based on (only) n. 4 animals that are MALES of a BEEF BREED, but all the reference used in the discussion, for comparison, is about DAIRY COWS. Justify this experimental approach

Results and discussion – Results are sometimes not clear. Discussion on conclusions do not reflect the importance given to aspects such as sustainability and form (i.e., pellet) of the product, and to the fact that it is ana alternative to more traditional protein sources. These aspects should be discussed more (e.g., with comparisons with other protein sources, feed forms, etc.). Therefore, ddiscussion must be improved. 

Conclusions Improve taking into account the previous comments. Furthermore, at the end of the conclusion, the target shifts on MILK production

References – Too much references from yourselves. Revise the references list according to the comments

 SPECIFIC COMMENTS

L. 46-75. They should be one paragraph or two separate paragraphs where the authors should organise information and use references better;

L. 80-81. Add more references

L. 86-89. The objective and hypothesis statements must be expanded and improved, including a comment on the pellet form of the feed.

L. 89-90. Delete

L. 99. kg (not kilograms)

L. 108-109. Not capital letters

L. 130-131. Which kind of protein concentrate? Add more details

L. 153 (paragraph 2.4) How did you determine nutrient digestibility? Add analytical details

L. 159. 20 °C (not 20°C)

L. 162 60 °C and so on (number °C)

Lines 164-165. Report the specific AOAC methods (i.e., #934.01 AOAC procedure for DM, and so on …)

L. 166. Add more detail about the  NDF (aNDF??), ADF and ADL lab methods (Ankom? Vogel et al., 1999?, alpha-amylase? bags?)  

L. 190-192. Add details for plasma analysis

L. 196. Report the equation of the model

L. 198-199. Which kind of orthogonal contrast …report the formula

L. 204-210. Move on M&M

L. 210-211. Delete it’s redundant and it’s too much speculative

L. 213-216. Delete it’s redundant and it’s too much speculative

L. 220 (p > 0.05)

L. 222 ‘discernible’? maybe “significant”

L. 225 (p < 0.05) … and so on across the manuscript

L. 267-322. Paragraphs 4.2 and 4.3 are repetitive and information must be reorganised.

L. 270-272. Not clear; there must be a mistake

L. 277-280: Misplaced. Moreover, not clear. Explain better

L. 319-320. Not clear; there must be a mistake

L. 324-327. Delete it’s redundant and it sounds as introduction

L. 331 add % to 6.81

L. 340 20 g

L. 340-344. Not clear. Rewrite since it’s no enough clear

L. 344-346. Delete it’s redundant and it sounds as introduction

L. 351-353. Improve the conclusions

L. 353-355. Delete

L. 351-355. Conclusions very poor, need to be improved and expanded.

Table 1

Report s.d. for the chemical composition

Tables 2, 3, 4 and 5

Explain control

Legend, write: ‘p < 0.05’

Comments on the Quality of English Language

Minor editing revision of English style

Author Response

Response to Reviewer 4:

All the manuscript – There are some style and English mistakes … check!

Response: Thank you for your comment and concern on this point. We made modifications based on your suggestions.

Title - shorten

Response: Thank you for your comment and we have shorted as “Feeding Pellets Containing Agro-industrial Waste Enhances Feed Utilization and Rumen Functions in Thai Beef Cattle” which was reduce word count by seven words.

Simple summary – Revise according the specific comments

Response: Thank you for your comment and concern on this point.

Abstract - Revise according the specific comments

Response: Thank you for your comment and concern on this point.

Keyword:  add ‘citric waste fermented’; ‘beef’ or ‘dairy cow’?

Response: Thank you for your comment and concern on this point. Thank you for your comment and concern on this point. We made modifications based on your suggestions.

Introduction - Revise according the specific comments

Response: Thank you for your comment and concern on this point.

M&M The experimental design is based on (only) n. 4 animals that are MALES of a BEEF BREED, but all the reference used in the discussion, for comparison, is about DAIRY COWS. Justify this experimental approach

Response: Thank you for your comment and concern on this point. We have incorporated citation regarding beef cattle into the section of discussion. Please see the manuscript.

Results and discussion – Results are sometimes not clear. Discussion on conclusions do not reflect the importance given to aspects such as sustainability and form (i.e., pellet) of the product, and to the fact that it is ana alternative to more traditional protein sources. These aspects should be discussed more (e.g., with comparisons with other protein sources, feed forms, etc.). Therefore, discussion must be improved.

Response: Thank you for your comment and concern on this point. We have done our best to improve the discussion of results. Please see in the manuscript.

Conclusions – Improve taking into account the previous comments. Furthermore, at the end of the conclusion, the target shifts on MILK production

Response: Thank you for your comment and concern on this point. The matter has been solved by removing of any relation to the production of milk.

References – Too much references from yourselves. Revise the references list according to the comments

Response: Thank you for your comment and concern on this point. We have already removed 3 self-references and remained only essential citation own work for 6 citations. Please see in manuscript.

SPECIFIC COMMENTS

  1. 46-75. They should be one paragraph or two separate paragraphs where the authors should organise information and use references better;

Response: Thank you for your comment and concern on this point. We made modifications based on your suggestions.

  1. 80-81. Add more references

Response Thank you for your comment and concern on this point. We made modifications based on your suggestions.

  1. 86-89. The objective and hypothesis statements must be expanded and improved, including a comment on the pellet form of the feed.

Response Thank you for your comment and concern on this point. We made modifications based on your suggestions as “It is hypothesized that the pelleting process has the potential to enhance feed quality, and the incorporation of pellet products may strategically improve rumen fermentation and optimize feed utilization in animals, thereby contributing to overall animal nutrition and performance. Consequently, this study aimed to explore the effects of strategic supplementation citric waste fermented with yeast waste pellet (CWYWP) on feed intake, rumen characteristics, and blood metabolites in Thai native beef cattle fed a rice straw-based diet.”

  1. 89-90. Delete

Response: Thank you for your comment and concern on this point. We made modifications based on your suggestions.

  1. 99. kg (not kilograms)

Response Thank you for your comment and concern on this point. We made modifications based on your suggestions.

  1. 108-109. Not capital letters

Response Thank you for your comment and concern on this point. We made modifications based on your suggestions.

  1. 130-131. Which kind of protein concentrate? Add more details

Response: Thank you for your comment and concern on this point. We made modifications based on your suggestions. “crude protein of concentrate diet”

  1. 153 (paragraph 2.4) How did you determine nutrient digestibility? Add analytical details

Response: Thank you for your comment and concern on this point. We made modifications based on your suggestions.  As follow:

Digestibility (%) = × 100% 

  1. 159. 20 °C (not 20°C)

Response: Thanks, we have added space as comment.

  1. 162 60 °C and so on (number °C)

Response: Thanks, we have added space as comment.

Lines 164-165. Report the specific AOAC methods (i.e., #934.01 AOAC procedure for DM, and so on …)

Response Thank you for your comment and concern on this point. We made modifications based on your suggestions as “The chemical composition, encompassing dry matter (DM; ID 967.03), ash (ID 942.05), and crude protein (CP; ID 984.13), was determined in accordance with the AOAC standard method [28].”. Please look at manuscript.

  1. 166. Add more detail about the NDF (aNDF??), ADF and ADL lab methods (Ankom? Vogel et al., 1999?, alpha-amylase? bags?)

Response Thank you for your comment and concern on this point. We made modifications based on your suggestions. Please look at manuscript.

  1. 190-192. Add details for plasma analysis

Response Thank you for your comment and concern on this point. We made modifications based on your suggestions. Please look at manuscript.

  1. 196. Report the equation of the model

Response: Thank you for your comment and concern on this point. We made modifications based on your suggestions.

  1. 198-199. Which kind of orthogonal contrast …report the formula

Response:

  1. 204-210. Move on M&M

Response Thank you for your comment and concern on this point. We made modifications based on your suggestions.

  1. 210-211. Delete it’s redundant and it’s too much speculative

Response Thank you for your comment and concern on this point. We made modifications based on your suggestions.

  1. 213-216. Delete it’s redundant and it’s too much speculative

Response Thank you for your comment and concern on this point. We made modifications based on your suggestions.

  1. 220 (p > 0.05)

Response Thank you for your comment and concern on this point. We made modifications based on your suggestions.

  1. 222 ‘discernible’? maybe “significant”

Response Thank you for your comment and concern on this point. We made modifications based on your suggestions.

  1. 225 (p < 0.05) … and so on across the manuscript

Response Thank you for your comment and concern on this point. We made modifications based on your suggestions.

  1. 267-322. Paragraphs 4.2 and 4.3 are repetitive and information must be reorganized.

Response Thank you for your comment and concern on this point. We made modifications based on your suggestions.

  1. 270-272. Not clear; there must be a mistake

Response Thank you for your comment and concern on this point. We discuss the mechanism by which increasing CP supplementation in the diet will have a beneficial effect on the microorganisms in the rumen.

  1. 277-280: Misplaced. Moreover, not clear. Explain better

Response Thank you for your comment and concern on this point. We rewrite, it involves compressing pellets to help bypass diets so it can be better digested and absorbed in the small intestine.

  1. 319-320. Not clear; there must be a mistake

Response Thank you for your comment and concern on this point. We delete this text already.

  1. 324-327. Delete it’s redundant and it sounds as introduction

Response: Thank you for your comment and concern on this point. We made modifications based on your suggestions.

  1. 331 add % to 6.81

Response: Thank you for your comment and concern on this point. We made modifications based on your suggestions.

  1. 340 20 g

Response: Thank you for your comment and concern on this point. We made modifications based on your suggestions.

  1. 340-344. Not clear. Rewrite since it’s no enough clear

Response: Thank you for your comment and concern on this point. We made modifications based on your suggestions.

  1. 344-346. Delete it’s redundant and it sounds as introduction

Response: Thank you for your comment and concern on this point. We made modifications based on your suggestions.

  1. 351-353. Improve the conclusions

Response: Thank you for your comment and concern on this point. We made modifications based on your suggestions.

  1. 353-355. Delete

Response: Thank you for your comment and concern on this point. We made modifications based on your suggestions.

  1. 351-355. Conclusions very poor, need to be improved and expanded.

Response: Thank you for your comment and concern on this point. We made modifications based on your suggestions.

Table 1

Report s.d. for the chemical composition

Response Thank you for your comment and concern on this point. We made modifications based on your suggestions.

Tables 2, 3, 4 and 5

Explain control

Response: Thank you for your comment and concern on this point. We have added.

Legend, write: ‘p < 0.05’

Response Thank you for your comment and concern on this point. We made modifications based on your suggestions.

Round 2

Reviewer 1 Report

Comments and Suggestions for Authors

Abstract The objective of this research was to investigate the effects of citric waste fermented with yeast waste pellet (CWYWP) supplementation on feed intake, rumen characteristics, and blood metabolites in Thai native beef cattle fed a rice straw-based diet. Four male Thai native beef cattle (1.0-1.5 years old) with an initial body weight (BW) of 116 ± 16 kg were utilized in a 4×4 Latin square design with 21-day periods. The animals were assigned to receive CWYWP supplementation at 0%, 2%, 4%, and 6% of the total dry matter (DM) intake per day. The results indicate that feeding beef cattle with CWYWP led to a linear increase in the intake of crude protein (CP) and the digestibility of CP, with the maximum levels observed at 6% CWYWP supplementation. Rumen characteristics, including pH, blood urea-nitrogen concentration, and protozoal population, showed no significant alterations in response to varying CWYWP dosages (p>0.05). However, CWYWP supplementation resulted in significant changes in the concentration of ammonia-nitrogen, with a notable increase observed when 4-6% CWYWP was introduced (p<0.05). The inclusion of 6% CWYWP resulted in the highest population of ruminal bacteria (p<0.05). Additionally, CWYWP supplementation led to a statistically significant increase in the mean propionic acid concentration compared to the group that did not receive CWYWP supplementation (p<0.05). In conclusion, this experiment demonstrates that supplementing Thai native beef cattle with CWYWP at either 4% or 6% DM per day can enhance total CP intake, CP digestibility, rumen bacterial population, and increase propionate concentration. If there was a difference in values, authors should use P < 0.05. The Latin square design should have more repetitions, therefore we recommend using a double QL

My observation regarding the number of replications used in Latin square designs refers to the periodic principles. In an experiment with four treatments, the use of a Double Latin Square design with eight animals, will increase the data analysis accuracy. 

Author Response

Response to Reviewer 1:

Abstract The objective of this research was to investigate the effects of citric waste fermented with yeast waste pellet (CWYWP) supplementation on feed intake, rumen characteristics, and blood metabolites in Thai native beef cattle fed a rice straw-based diet. Four male Thai native beef cattle (1.0-1.5 years old) with an initial body weight (BW) of 116 ± 16 kg were utilized in a 4×4 Latin square design with 21-day periods. The animals were assigned to receive CWYWP supplementation at 0%, 2%, 4%, and 6% of the total dry matter (DM) intake per day. The results indicate that feeding beef cattle with CWYWP led to a linear increase in the intake of crude protein (CP) and the digestibility of CP, with the maximum levels observed at 6% CWYWP supplementation. Rumen characteristics, including pH, blood urea-nitrogen concentration, and protozoal population, showed no significant alterations in response to varying CWYWP dosages (p>0.05). However, CWYWP supplementation resulted in significant changes in the concentration of ammonia-nitrogen, with a notable increase observed when 4-6% CWYWP was introduced (p<0.05). The inclusion of 6% CWYWP resulted in the highest population of ruminal bacteria (p<0.05). Additionally, CWYWP supplementation led to a statistically significant increase in the mean propionic acid concentration compared to the group that did not receive CWYWP supplementation (p<0.05). In conclusion, this experiment demonstrates that supplementing Thai native beef cattle with CWYWP at either 4% or 6% DM per day can enhance total CP intake, CP digestibility, rumen bacterial population, and increase propionate concentration. If there was a difference in values, authors should use P < 0.05. The Latin square design should have more repetitions, therefore we recommend using a double QL

My observation regarding the number of replications used in Latin square designs refers to the periodic principles. In an experiment with four treatments, the use of a Double Latin Square design with eight animals, will increase the data analysis accuracy.

Response: We appreciate the comments made by Reviewer 1, and we agree that increasing the number of animal experiments, specifically using a Double Latin Square design with eight animals, would enhance the accuracy of data analysis. However, due to limitations in the number of animals available in our department's farm (with the same range of body weight, age, and other conditions), we cannot include additional animals in the present study. Furthermore, despite using a lower number of animals, we have made significant efforts to address the limitations of this condition. These efforts include avoiding errors during data collection, practicing with animals for feeding and pen-related activities before conducting experiments, and training personnel to properly take samples. These measures contribute to ensuring data accuracy and accountability.

In any case, we will consider your comments for future studies and plan to explore the use of a Double Latin Square design with eight animals as a replacement for the single Latin Square design. We appreciate your valuable comments and suggestions. Thank you again.

Reviewer 2 Report

Comments and Suggestions for Authors

The authors have made important improvements to the manuscript. However, my original criticism and reason for not recommending publication remains. The Latin square design is an elegant means of improving the statistical power when resources limit the ability to conduct more extensive experimentation. Yet, the high standard errors are problematic. For example, with only four reps per treatment a 14% increase in DM intake in non-significant. Under practical conditions, a 14% increase in intake could increase liveweight gain by a substantial amount with positive economic consequences for the farmer.  The precision of your design is not enough to test the null hypothesis and you make a type II error: you fail to reject the null hypothesis when it is false. Therefore, regrettably I stand by my original view that this paper is not suitable for publication.

Comments on the Quality of English Language

Some minor grammatical errors to be resolved.

Author Response

Response to Reviewer 2:

The authors have made important improvements to the manuscript. However, my original criticism and reason for not recommending publication remains. The Latin square design is an elegant means of improving the statistical power when resources limit the ability to conduct more extensive experimentation. Yet, the high standard errors are problematic. For example, with only four reps per treatment a 14% increase in DM intake in non-significant. Under practical conditions, a 14% increase in intake could increase liveweight gain by a substantial amount with positive economic consequences for the farmer.  The precision of your design is not enough to test the null hypothesis and you make a type II error: you fail to reject the null hypothesis when it is false. Therefore, regrettably I stand by my original view that this paper is not suitable for publication.

Response: We once again appreciate your opinion on the revised version of our manuscript, and your point is quite clear. Consequently, we made a concerted effort to re-examine the raw data. After carefully reviewing the raw data and conducting a re-statistical analysis, we observed slight changes in the data presented in Table 2 (feed and nutrient intake).

We identified a significant difference in total feed intake with a low standard error of the mean (SEM). This result has implications for the number of nutrient intakes, although the differences were not statistically significant, except for crude protein (CP) intake. The details of the data re-analysis are demonstrated in Table 2, and the sections on results and discussion have been modified accordingly.

We trust that the revised version of the manuscript will bridge the gap between the use of a low number of animals in the Latin square design and the accuracy and accountability of the statistical data. Thank you for your kind consideration of the manuscript for publication.

Reviewer 3 Report

Comments and Suggestions for Authors

Manuscript is improved and can be published in present form.

Author Response

Response to Reviewer 3:

Manuscript is improved and can be published in present form.

Response: We appreciate your previous comment and recently recommend the paper for acceptance for publication. We believe that if the present work is published, it might be useful to farmers who would like to incorporate the residue into animal feed, thereby reducing environmental impact. Thank you for your kind consideration of the manuscript for publication.

Reviewer 4 Report

Comments and Suggestions for Authors

No further suggestions for the Authors

Comments on the Quality of English Language

A minor English revision should be done.

Author Response

Response to Reviewer 4:

No further suggestions for the Authors

Response: We appreciate your previous comment and recently recommend the paper for acceptance for publication. We believe that if the present work is published, it might be useful to farmers who would like to incorporate the residue into animal feed, thereby reducing environmental impact. Thank you for your kind consideration of the manuscript for publication.